# OpenReview forum: "Improving Generalization of Alignment with Human Preferences through Group Invariant Learning"
_ICLR.cc/2024/Conference — ICLR 2024 spotlight_

### Official Review · Reviewer_cSKU · 2023-10-29

**Soundness:** 3 good
**Presentation:** 3 good
**Contribution:** 2 fair
**Rating:** 8
**Confidence:** 4

**Summary:**

This paper suggests a method for Improving RLHF by preventing it to learn shortcuts/reward-hacking using a concept data groups where groups are classified rather than given beforehand. At the training time, the suggested approach puts more focus on learning the best behavior for challenging groups.

The grouping objective is adversarial to find groups with maximum discrepancy in reward, while the RL objective is encouraging reduction of loss variance across groups

-- Updated my evaluation after reading the responses and revisions

**Strengths:**

- The paper is well written and clear
- The method is generally sound and intuitive
- Promising experiment results

**Weaknesses:**

1- It is unclear how to find the optimal number of groups? The paper seems to miss discussion of this and sharing what settings used in the presented results, and how authors reached that setting.

2- The adversarial objective may pickup noisy or outliers and divert the optimization. I think this deserves more investigations.

3- The learning objective is encouraging reduction of loss variance across groups, I was wondering why authors didn’t directly go for optimization macro average of loss which is a bit more intuitive and has less chance to have side-effects on the overall performance as mean is generally more stable than variance?

4- Robust optimization has been extensively studied in the general optimization context. Many of such methods could be applicable to the RLHF/LLM problem and the method proposed in this paper is also also applicable to other settings. I do not see any comparisons to support this method is optimal for RLHF compared to existing robust optimization methods.

5- Related to the previous two comments, we we should have compared with other robust optimization methods and variations of applying the group loss

6- In Section 3, “The training of an AI assistant consists of three main stages...“ is not necessarily the case for all AI Assistants. I suggest revising this statement and connecting the presented method to more broader usecases as it is not really limited to this case.

**Questions:**

(see points mentioned above)

**Details Of Ethics Concerns:**

No particular concern

---

> ### Author Response · Authors · 2023-11-21
> **Official Comment by Authors**
>
> Thank you to the reviewer for providing insightful comments. We have addressed each of your suggestions and incorporated additional experiments accordingly. For example, in **Appendix B.3**, we have included an analysis of the **effect of group size**, and in **Appendix B.4**, we have added a **comparison with other robust optimization methods**. If you have any further comments, please do let us know, and we will do our best to address them.
>
> **Q1: It is unclear how to find the optimal number of groups? The paper seems to miss discussion of this and sharing what settings used in the presented results, and how authors reached that setting.**
>
> A1: Thank you to the reviewer for pointing out this issue. In the previous version of the manuscript, in section 4.3, we introduced that all of our experiments were conducted with two groups. We followed the reviewer's advice and conducted further experiments to investigate the impact of group size on the performance of our method.We first show the performance of our method under various group sizes.
>
> | Group  Size | HH-Harmful |      |       | HH-Helpful |      |       | OpenAI-Summary |      |       |
> | ----------- | ---------- | ---- | ----- | ---------- | :--- | ----- | -------------- | ---- | ----- |
> |             | Win↑       | Tie  | Lose↓ | Win↑       | Tie  | Lose↓ | Win↑           | Tie  | Lose↓ |
> | 2           | 40.4       | 33.7 | 25.9  | 29.5       | 63.8 | 6.7   | 34.1           | 48.2 | 17.7  |
> | 3           | 46.6       | 22.8 | 19.6  | 34.2       | 57.6 | 8.2   | 33.7           | 47.3 | 19.0  |
> | 5           | 47.6       | 31.9 | 20.5  | 35.1       | 53.1 | 11.8  | 27.6           | 55.4 | 17.0  |
> | 10          | 47.6       | 36.2 | 16.2  | 35.1       | 53.5 | 11.4  | 33.8           | 40.5 | 26.1  |
>
> From the above results, we can draw the following conclusions:
>
> 1) When using only two groups, we can achieve satisfactory performance [1, 2, 3]. Prior researches have also demonstrated this, and hence, in our initial experimental setup, we followed this simple setting.
> 2)  When the group size is greater than $3$, increasing the group size has a relatively minor impact on model performance. This conclusion aligns with the findings in the ZIN [4], where they discovered that when the group size is greater than or equal to $4$, the impact of group size on performance is relatively small. This finding suggests that once the group size is increased beyond a certain threshold, there is no need to further increase the group size. Therefore, in practical applications, if it is difficult to determine the optimal group size, opting for a slightly larger group size can be a reasonable choice.
>
> These advantages are attributed to our method's **dynamic data division** based on performance metrics, ensuring that even in multi-group scenarios, our model can identify and optimize for the worst-performing group. This approach allows us to maintain focus on the most challenging aspects of the data, ensuring effective optimization regardless of the number of groups present.
>
> We **have included** the above analysis and results in **Appendix B.3**.
>
>
>
> **Q2: The adversarial objective may pickup noisy or outliers and divert the optimization. I think this deserves more investigations.**
>
> A2: Thank you for raising this concern. Our group labeling inference method **dynamically operates** on **each batch**, minimizing the risk of being influenced by noisy data or outliers. By continuously updating group labels in each new batch, the model is less likely to focus on outlier data points. Instead, it adapts to more general data patterns that emerge throughout the entire training process.
>
> In the context of invariant learning and distributionally robust optimization (DRO), the methodology inherently aims to learn representations that are stable across various distributions, thereby providing a **natural defense against the influence of outliers**. As elucidated by Arjovsky et al. (2019) in their work on Invariant Risk Minimization [1], models trained to discover invariant features across different environments can generalize better to unseen scenarios. This is because invariant features, by definition, are those that do not depend on the specificities of a single data distribution and hence are less susceptible to noise and outliers.
>
> Furthermore, DRO approaches [2], involve optimizing for the **worst-case distribution** within a certain ambiguity set, which typically leads to solutions that are less sensitive to data perturbations, including noise and outliers.
>
> In our case, the **dynamic nature of group label inference** across batches can be likened to a form of online DRO, where the model is constantly being challenged to perform well across a series of worst-case scenarios defined by each new batch. This process lead to a model that is robust to the influence of individual data points.

---

> ### Author Response · Authors · 2023-11-21
> **Official Comment by Authors**
>
> **Q3:  The learning objective is encouraging reduction of loss variance across groups, I was wondering why authors didn’t directly go for optimization macro average of loss which is a bit more intuitive and has less chance to have side-effects on the overall performance as mean is generally more stable than variance?**
>
> A3: Thank you for your insightful question!
>
> Our decision to target loss variance stems from the **core principles of invariant learning**, which seeks to identify and leverage stable patterns across different environments or subpopulations [1]. Invariant learning is fundamentally about ensuring that the learned model's performance is not overly dependent on specific attributes or conditions of the data that may vary between groups. By minimizing variance, we aim to ensure that our model's performance is consistent across groups, which is more aligned with the concept of invariance than optimizing for the macro loss.
>
> We conducted experiments to validate the performance of the proposed method compared to the method based on marco average loss. The experimental results are shown in the table below.
>
> | Methods | HH-Harmful |      |       | HH-Helpful |      |       | OpenAI-Summary |      |       |
> | ------- | ---------- | ---- | ----- | ---------- | ---- | ----- | -------------- | ---- | ----- |
> |         | Win↑       | Tie  | Lose↓ | Win↑       | Tie  | Lose↓ | Win↑           | Tie  | Lose↓ |
> | Ours    | 40.4       | 33.7 | 25.9  | 29.5       | 63.8 | 6.8   | 34.1           | 48.2 | 17.7  |
> | Macro   | 26.6       | 45.3 | 28.1  | 15.5       | 54.4 | 30.2  | 19.3           | 39.4 | 41.2  |
>
>  The performance of the marco average loss is worse than that of the variance and is also weaker than PPO. This is because optimizing for **marco average loss** could still **allow for significant performance discrepancies across groups**, further aggravating reward hacking. This is because reward hacking still aligns with the optimization objective of the marco average loss. In contrast, minimizing variance directly addresses the disparities, compelling the model to perform consistently well across all groups. This consistently is critical for models expected to operate in diverse and unpredictable real-world settings, where the ability to generalize across different scenarios is paramount.
>
> We **have included** the above analysis and results in **Appendix B.4**.

---

> ### Author Response · Authors · 2023-11-21
> **Official Comment by Authors**
>
> **Q4: Robust optimization has been extensively studied in the general optimization context. Many of such methods could be applicable to the RLHF/LLM problem and the method proposed in this paper is also also applicable to other settings. I do not see any comparisons to support this method is optimal for RLHF compared to existing robust optimization methods.**
>
> A4: Thank you for your comment and the opportunity to further show our novelty and advantages over other robust optimization methods.
>
> Robust optimization has been extensively studied in the general optimization context. Given that our approach and previous methods address similar concerns—how to ensure that a model exhibits consistent performance across different data distributions and possesses strong generalization abilities—it is imperative to compare our approach with prior robust optimization techniques. However, previous robust optimization methods **cannot be directly applied** to our RLHF pipeline, primarily due to two reasons:
>
> 1. Our primary motivation arises from observing inconsistent performance across different data samples when using PPO in RLHF pipline, whereas previous methods primarily focus on classification tasks.
>
> 2. it is challenging to manually label the dataset with group labels because the RLHF training process is dynamic. Our approach dynamically conducts group inference during the training process, whereas previous group inference methods employ a two-stage approach: first, training on the entire dataset, and then training a classifier for group inference.
>
> In the design phase of our method, we conducted a cross-comparison with methods such as Group DRO [3] and JTT [4]. Once group labels are provided using our proposed method, we can adopt optimization objectives similar to those in Group DRO and JTT. Experimental results are as follows:
>
> | Methods   | HH-Harmful |      |       | HH-Helpful |      |       | OpenAI-Summary |      |       |
> | --------- | ---------- | ---- | ----- | ---------- | ---- | ----- | -------------- | ---- | ----- |
> |           | Win↑       | Tie  | Lose↓ | Win↑       | Tie  | Lose↓ | Win↑           | Tie  | Lose↓ |
> | Ours      | 40.4       | 33.7 | 25.9  | 29.5       | 63.8 | 6.7   | 34.1           | 48.2 | 17.7  |
> | Group DRO | 31.0       | 47.3 | 21.7  | 20.9       | 65.5 | 13.6  | 25.9           | 50.3 | 22.8  |
> | JTT       | 32.1       | 47.8 | 20.1  | 23.4       | 66.0 | 10.6  | 27.9           | 48.9 | 23.2  |
>
> As can be seen, methods based on **Group DRO and JTT outperform PPO** because they take into account further optimization for the worst group. At the same time, **our method outperforms both of these approaches**, possibly because our group label inference objective and policy invariant learning constitute an adversarial objective that can maximize the performance of the policy model. Additionally, this validates the **scalability of our proposed method**, providing the initial step towards applying robust optimization to RLHF.
>
> We **have included** the above results and more details in **Appendix B.4**.
>
>
>
> **Q5: Related to the previous two comments, we should have compared with other robust optimization methods and variations of applying the group loss.**
>
> A5: Thank you for the reviewer's feedback. We have conducted relevant experiments; please refer to A3 and A4 for details.  **If you have any questions, please let us know, and we will do our best to address your concerns.**

---

> ### Author Response · Authors · 2023-11-21
> **Official Comment by Authors**
>
> **Q6: In Section 3, “The training of an AI assistant consists of three main stages...“ is not necessarily the case for all AI Assistants. I suggest revising this statement and connecting the presented method to more broader use cases as it is not really limited to this case.**
>
> A6: Thank you for pointing it out. We have revised this statement as follows to provide a better introduction to the RLHF pipeline:
>
> We review the RLHF pipeline from [5], which has been applied to tasks like dialogue [6], instruction following [7], and summarization [8]]. This pipeline typically includes three phases: supervised fine-tuning (SFT), preference sampling and reward model (RM) training, and RL fine-tuning using proximal policy optimization (PPO). The process usually starts with a generic pre-trained language model, which undergoes supervised learning on a high-quality dataset for specific downstream tasks, resulting in a model denoted as $\pi^\text{SFT}$.
>
> To more accurately reflect this in the manuscript, we **have revised** this statement.
>
> ---
> [1] Arjovsky, M., Bottou, L., Gulrajani, I., & Lopez-Paz, D. (2019). Invariant Risk Minimization. arXiv:1907.02893.
>
> [2] Rahimian, H., & Mehrotra, S. (2019). Distributionally Robust Optimization: A Review. arXiv:1908.05659.
>
> [3] Sagawa, S., Koh, P. W., Hashimoto, T. B., & Liang, P. (2019). Distributionally robust neural networks for group shifts: On the importance of regularization for worst-case generalization. arXiv preprint arXiv:1911.08731.
>
> [4] Liu, Evan Z., et al. "Just train twice: Improving group robustness without training group information." International Conference on Machine Learning. PMLR, 2021.
>
> [5] Ziegler, Daniel M., et al. "Fine-tuning language models from human preferences." arXiv preprint arXiv:1909.08593 (2019).
>
> [6] Glaese, Amelia, et al. "Improving alignment of dialogue agents via targeted human judgements." arXiv preprint arXiv:2209.14375 (2022).
>
> [7] Ouyang, Long, et al. "Training language models to follow instructions with human feedback." Advances in Neural Information Processing Systems 35 (2022): 27730-27744.
>
> [8] Stiennon, Nisan, et al. "Learning to summarize with human feedback." Advances in Neural Information Processing Systems 33 (2020): 3008-3021.

---

### Official Review · Reviewer_7D9U · 2023-11-01

**Soundness:** 4 excellent
**Presentation:** 4 excellent
**Contribution:** 4 excellent
**Rating:** 10
**Confidence:** 4

**Summary:**

The paper introduces a method for improving LLMs' response quality on rare or difficult examples in the training data. Rather than treating all examples as equally important, the paper proposes to separate the data into groups so as to maximize the variance in quality among groups, then train the model to minimize that variance so that the model performs at roughly the same quality for all groups. Essentially this automatically adds additional training weight to difficult examples. The paper then presents empirical evidence that the proposed approach substantially improves response quality.

**Strengths:**

The paper is well motivated, proposes a novel idea, describes it clearly, and presents compelling evidence that it works. In short, this is an excellent paper!

It also has immediate practical significance for training LLMs to produce higher quality outputs, and initial evidence suggests this would be more helpful and less harmful for users.

Figure 1 is an excellent visualization of the problem and the proposed solution.

Figure 4 clearly demonstrates that the proposed method is much more stable than vanilla PPO.

**Weaknesses:**

My concerns are minor.

- Sec 4.3 para 1: "the changes *to which* the algorithm needs to remain invariant..."
- p6, Stage 2:
    - "binary groups": Is there an advantage to using just 2 groups? I would have expected more. Variance is really just a simple difference unless there are more than two groups
    - "narrows the performance gap between the two groups": Relative to PPO, yes, but it still stays flat or increases over time. Why is that?
- p7, Table 1: It would be helpful to know how statistically significant these results are. Is there some way to do that? It would also be helpful to add a calibration where a method is evaluated against itself. It should be a 100% tie, but obviously it will be more like 33/34/33.
- p9, ablation study: "it enables the model to explore a larger action space on challenging samples, leading to the discovery of a superior policy". How does it enable that?
- p9, reward distribution: "which will aid in its generalization to unseen data." Why will it do that? Did you show this somewhere?

**Questions:**

I don't have any major questions. The authors can feel free to respond to my questions in the previous section if they have time.

---

> ### Author Response · Authors · 2023-11-21
> **Official Comment by Authors**
>
> Thank you very much for your recognition of our work!  We have addressed each of the issues you raised and made the necessary revisions to our manuscript. In Appendix B.1, we have added a statistical significance analysis, and in Appendix B.3, we have included an analysis of the effect of group size. **The changes are marked in blue.**
>
> **Q1: Sec 4.3 para 1: "the changes *to which* the algorithm needs to remain invariant..."**
>
>  A1: Thank you for pointing it out. In this context, "the algorithm" refers to the algorithm we are currently training in. In response, we **have revised** the relevant section accordingly.
>
>
>
> **Q2.1: "binary groups": Is there an advantage to using just 2 groups? I would have expected more. Variance is really just a simple difference unless there are more than two groups**
>
> A2.1: Yes, our method performs well when there are only two groups, and we further conducted our experiments to investigate the impact of larger group sizes on performance. We first show the performance of our method under various group sizes.
>
> | Group  Size | HH-Harmful |      |       | HH-Helpful |      |       | OpenAI-Summary |      |       |
> | ----------- | ---------- | ---- | ----- | ---------- | :--- | ----- | -------------- | ---- | ----- |
> |             | Win↑       | Tie  | Lose↓ | Win↑       | Tie  | Lose↓ | Win↑           | Tie  | Lose↓ |
> | 2           | 40.4       | 33.7 | 25.9  | 29.5       | 63.8 | 6.7   | 34.1           | 48.2 | 17.7  |
> | 3           | 46.6       | 22.8 | 19.6  | 34.2       | 57.6 | 8.2   | 33.7           | 47.3 | 19.0  |
> | 5           | 47.6       | 31.9 | 20.5  | 35.1       | 53.1 | 11.8  | 27.6           | 55.4 | 17.0  |
> | 10          | 47.6       | 36.2 | 16.2  | 35.1       | 53.5 | 11.4  | 33.8           | 40.5 | 26.1  |
>
> From the above results, we can draw the following conclusions:
>
> 1) When using only two groups, we can achieve satisfactory performance [1, 2, 3]. Prior researches have also demonstrated this, and hence, in our initial experimental setup, we followed this simple setting.
> 2)  When the group size is greater than $3$, increasing the group size has a relatively minor impact on model performance. This conclusion aligns with the findings in the ZIN [4], where they discovered that when the group size is greater than or equal to $4$, the impact of group size on performance is relatively small. This finding suggests that once the group size is increased beyond a certain threshold, there is no need to further increase the group size. Therefore, in practical applications, if it is difficult to determine the optimal group size, opting for a slightly larger group size can be a reasonable choice.
>
> These advantages are attributed to our method's **dynamic data division** based on performance metrics, ensuring that even in multi-group scenarios, our model can identify and optimize for the worst-performing group. This approach allows us to maintain focus on the most challenging aspects of the data, ensuring effective optimization regardless of the number of groups present.
>
> We **have included** the above analysis and results in **Appendix B.3**.
>
>
>
> **Q2.2: "narrows the performance gap between the two groups": Relative to PPO, yes, but it still stays flat or increases over time. Why is that?**
>
> A2.2: That's a very good question! The persistence of a performance gap can be due to several factors:
>
> 1) **Complexity of the Environment**: Different groups may represent varying levels of complexity within the environment, which inherently leads to different performance ceilings.
> 2) **Dynamic Environments**: As the environment evolves, the model continuously learns and adapts. The observed performance gap may reflect the model's current focus on learning more robust and invariant features, which are beneficial for generalization, rather than equalizing short-term rewards.
>
> I must admit that it's challenging for us to find an optimal strategy to completely eliminate performance gap among different groups. However, that is precisely our goal.

---

> ### Author Response · Authors · 2023-11-21
> **Official Comment by Authors**
>
> **Q3: p7, Table 1: It would be helpful to know how statistically significant these results are. Is there some way to do that? It would also be helpful to add a calibration where a method is evaluated against itself. It should be a 100% tie, but obviously it will be more like 33/34/33.**
>
> A3: Thank you for your insightful inquiries regarding the statistical significance of our results and the suggestion for calibration by evaluating a method against itself.
>
> To address the statistical significance, we conduct **Chi-Squared test** [5] to examine the significance of the performance differences between our proposed method and various baseline methods. It is difficult to calibrate a method when it is evaluated against itself to achieve a 100% tie, because neither the human evaluator nor the GPT-4 evaluator can provide the probability for each answer. Given the categorical nature of our data (win, tie, lose), and the assumption of equal performance among methods translating into a 33/34/33 ratio for these categories, the Chi-Squared test is an appropriate method to apply.
>
> | Opponent  | Chi2 Statistic | p-value  | Degrees of Freedom |
> | --------- | -------------- | -------- | ------------------ |
> | SFT       | 67.83          | 1.87e-15 | 2                  |
> | PPO       | 107.44         | 4.68e-24 | 2                  |
> | PPO w/ KL | 8.14           | 0.017    | 2                  |
> | DPO       | 8.54           | 0.014    | 2                  |
>
> The results of our Chi-Squared test, as shown above, indicate a **significant difference** in the win/tie/lose ratios compared to the expected 1:1:1 ratio. Specifically, the methods SFT and PPO showed highly significant deviations from the expected distribution, with p-values far below the **commonly accepted threshold of 0.05**. This suggests that our method has a statistically significant difference in performance compared to these baselines. Meanwhile, the methods PPO w/KL and DPO also exhibited significant results, but to a lesser extent.
>
> In conclusion, the statistical analysis confirms the superiority of our method over the compared baselines, with the consistency of results between GPT-4 and human evaluations further validating our approach.
>
> We **have included** the above analysis and results in **Appendix B.1**. We believe these additional analyses will greatly strengthen the validity and reliability of our results, and we appreciate your suggestions that led to these improvements.
>
>
> **Q4: p9, ablation study: "it enables the model to explore a larger action space on challenging samples, leading to the discovery of a superior policy". How does it enable that?**
>
> A4: We apologize that our statement is not clear enough.
>
> The dynamic KL penalty is designed to adjust the penalty strength based on the performance of the policy on different groups of data. For samples that are already performing well, the penalty is increased to discourage deviation from the learned policy. Conversely, for **challenging samples** where the performance is not yet optimal, the **penalty is reduced**, allowing the model to **explore actions that deviate more significantly from the current policy**. This increased exploration is critical in challenging regions of the state space, where a more extensive search is necessary to identify actions that may lead to better performance.
>
> To more accurately reflect this in the manuscript, we **have revised** this statement.

---

> > ### Comment · Reviewer_7D9U · 2023-11-22
> > **Thanks for your response**
> >
> > Thank you for the response, and for including the statistical significance test. This is really nice work.

---

> ### Author Response · Authors · 2023-11-21
> **Official Comment by Authors**
>
> **Q5: p9, reward distribution: "which will aid in its generalization to unseen data." Why will it do that? Did you show this somewhere?**
>
> A5: Thank you for your question regarding the connection between consistent performance across various samples and the generalization capabilities of our model to unseen data.
>
> The reward distribution that closely approximates a **Gaussian distribution** suggests that our model, trained with group invariant learning, is **less prone to overfitting to specific features or patterns** in the training data that do not generalize well. In invariant learning, the focus is on learning features and patterns that remain stable across different data distributions, which is a hallmark of causal relationships within the data. By encouraging the model to learn such **invariant features**, we aim to enhance its ability to generalize, as these features are **more likely to be present in out-of-distribution (OOD) data** and hence contribute to more reliable predictions [6].
>
> In contrast, the long-tailed distribution observed with PPO suggests that while it may achieve higher rewards on some training samples, it is likely exploiting non-invariant features that might not generalize beyond the training distribution. This can lead to less robust performance on OOD data.
>
> To clarify this in the manuscript, we **have revised** the relevent text.
>
> ---
>
> [1]  Arjovsky, M., Bottou, L., Gulrajani, I., & Lopez-Paz, D. (2019). Invariant risk minimization. arXiv preprint arXiv:1907.02893.
>
> [2] Creager, E., Jacobsen, J. H., & Zemel, R. (2021, July). Environment inference for invariant learning. In International Conference on Machine Learning (pp. 2189-2200). PMLR.
>
> [3] Chen, Y., Xiong, R., Ma, Z. M., & Lan, Y. (2022). When Does Group Invariant Learning Survive Spurious Correlations?. *Advances in Neural Information Processing Systems*, *35*, 7038-7051.
>
> [4] Lin, Y., Zhu, S., Tan, L., & Cui, P. (2022). ZIN: When and How to Learn Invariance Without Environment Partition?. *Advances in Neural Information Processing Systems*, *35*, 24529-24542.
>
> [5] Pearson, K. (1900). X. On the criterion that a given system of deviations from the probable in the case of a correlated system of variables is such that it can be reasonably supposed to have arisen from random sampling. *The London, Edinburgh, and Dublin Philosophical Magazine and Journal of Science*, *50*(302), 157-175.
>
> [6]  Arjovsky, M., Bottou, L., Gulrajani, I., & Lopez-Paz, D. (2019). Invariant risk minimization. arXiv preprint arXiv:1907.02893.

---

### Official Review · Reviewer_bjuC · 2023-11-02

**Soundness:** 2 fair
**Presentation:** 3 good
**Contribution:** 3 good
**Rating:** 6
**Confidence:** 3

**Summary:**

The paper is motivated by the notion that maximizing only the expected return of a policy can be suboptimal when the return distribution has high variance or long tails. In Reinforcement Learning, the goal is to learn a policy that performs consistently across different data groups or scenarios. Here the group refers to different data distributions with varying performance characteristics. To achieve this, the authors propose incorporating a group inference classifier in the critic model to softly assign data points to groups. They maximize a variance objective to amplify the differences between groups. Additionally, they introduce an adaptive KL regularization scheme that allows different groups to have varied regularization strengths. This balances exploration and exploitation based on the group's difficulty. In summary, the paper aims to improve policy generalization by minimizing performance disparities between automatically identified data groups, while dynamically adapting the regularization to enable optimized exploration.

**Strengths:**

pros:
1. This paper introduced a framework for policy invariant learning that does not reply on prior domain or group knowledge, where labeling is inefficient.
2. They propose a novel dynamic KL penalty based on group labels discovered before. Easier groups get stricter constraints, potentially preventing overoptimization.
3. Good experimental results. 	The method demonstrates good experimental results, outperforming baselines on in-distribution data. This highlights its capabilities in known domains. On out-of-distribution data, it shows even greater performance gains, underscoring its generalization abilities.
4. Ablation studies validate that both core components - group invariant learning and adaptive KL - contribute to the overall performance gains.

**Weaknesses:**

Cons:
1. The experiments seem to imply a static number of groups, binary in the experiments,(best and challenging groups) which is not ideal with multi-modal data in dynamics environments. Experimenting with varying number of groups would be better to see how well does the method perform when the number of data groups increases?
2. The approach relies on accurate group label inference. Incorrect initial grouping could potentially lead to improper KL penalties. How robust is the method to errors in initial group assignment?
3. question: how do the newly introduced hyperparameters add complexity to tuning the algorithm? as training PPO is already known to be hard to tune.

**Questions:**

please see weakness

---

> ### Author Response · Authors · 2023-11-21
> **Official Comment by Authors**
>
> Thank you for your valuable comments! In accordance with your suggestion, we have included **sensitivity experiments for hyperparameters** and the analysis of the **effect of group size** in **Appendix B.2** and **Appendix B.3**, respectively. **The changes are marked in blue.**
>
> **Q1: The experiments seem to imply a static number of groups, binary in the experiments,(best and challenging groups) which is not ideal with multi-modal data in dynamics environments. Experimenting with varying number of groups would be better to see how well does the method perform when the number of data groups increases?**
>
> A1: Thank you for your valuable feedback regarding the use of a static number of groups in our experiments. We first show the performance of our method under various group sizes.
>
> | Group  Size | HH-Harmful |      |       | HH-Helpful |      |       | OpenAI-Summary |      |       |
> | ----------- | ---------- | ---- | ----- | ---------- | :--- | ----- | -------------- | ---- | ----- |
> |             | Win↑       | Tie  | Lose↓ | Win↑       | Tie  | Lose↓ | Win↑           | Tie  | Lose↓ |
> | 2           | 40.4       | 33.7 | 25.9  | 29.5       | 63.8 | 6.7   | 34.1           | 48.2 | 17.7  |
> | 3           | 46.6       | 22.8 | 19.6  | 34.2       | 57.6 | 8.2   | 33.7           | 47.3 | 19.0  |
> | 5           | 47.6       | 31.9 | 20.5  | 35.1       | 53.1 | 11.8  | 27.6           | 55.4 | 17.0  |
> | 10          | 47.6       | 36.2 | 16.2  | 35.1       | 53.5 | 11.4  | 33.8           | 40.5 | 26.1  |
>
> From the above results, we can draw the following conclusions:
>
> 1) When using only two groups, we can achieve satisfactory performance [1, 2, 3]. Prior researches have also demonstrated this, and hence, in our initial experimental setup, we followed this simple setting.
> 2)  When the group size is greater than $3$, increasing the group size has a relatively minor impact on model performance. This conclusion aligns with the findings in the ZIN [4], where they discovered that when the group size is greater than or equal to $4$, the impact of group size on performance is relatively small. This finding suggests that once the group size is increased beyond a certain threshold, there is no need to further increase the group size. Therefore, in practical applications, if it is difficult to determine the optimal group size, opting for a slightly larger group size can be a reasonable choice.
>
> These advantages are attributed to our method's **dynamic data division** based on performance metrics, ensuring that even in multi-group scenarios, our model can identify and optimize for the worst-performing group. This approach allows us to maintain focus on the most challenging aspects of the data, ensuring effective optimization regardless of the number of groups present.
>
> We **have included** the above analysis and results in **Appendix B.3**.
>
>
>
> **Q2: The approach relies on accurate group label inference. Incorrect initial grouping could potentially lead to improper KL penalties. How robust is the method to errors in initial group assignment?**
>
> A2: Thank you for your question regarding the robustness of our method to errors in initial group label assignment. In our approach, the classifier used for group label inference is **initialized randomly**, meaning the **initial distribution does not contain any prior information**. This design choice is critical as it allows the method to learn and adapt based on the data, without any biases introduced by preconceived groupings.
>
> Our method leverages gradient descent to iteratively refine and accurately infer group labels. This adaptive learning process is key to the robustness of our approach. Even if the initial grouping is not accurate, the method is designed to correct and improve the group assignments through successive iterations.
>
> Furthermore, our extensive experimental results demonstrate the robustness of this approach. We have observed consistent performance across a variety of scenarios, indicating that our method effectively learns and adapts to accurate group label inference, thereby mitigating the risks associated with incorrect initial groupings.

---

> ### Author Response · Authors · 2023-11-21
> **Official Comment by Authors**
>
> **Q3: How do the newly introduced hyperparameters add complexity to tuning the algorithm? as training PPO is already known to be hard to tune.**
>
> A3: Thank you for pointing it out. Because PPO is already known to be challenging to tune, it is essential for us to analyze the sensitivity of our method to newly introduced parameters.
> In practical applications, we fix hyperparameter $\beta_{\mathrm{critic}}$ to $1$, as it primarily controls the learning of the group inference classifier without affecting the critic model itself. Our parameter tuning is mainly focused on hyperparameter $\beta_{\mathrm{policy}}$.
>
> | β_policy | HH-Harmful |      |       | HH-Helpful |      |       | OpenAI-Summary |      |       |
> | -------- | ---------- | ---- | ----- | ---------- | :--- | ----- | -------------- | ---- | ----- |
> |          | Win↑       | Tie  | Lose↓ | Win↑       | Tie  | Lose↓ | Win↑           | Tie  | Lose↓ |
> | 0.01     | 40.4       | 33.7 | 25.9  | 29.5       | 63.8 | 6.7   | 34.1           | 48.2 | 17.7  |
> | 0.05     | 34.4       | 41.2 | 24.4  | 27.1       | 64.2 | 8.6   | 28.9           | 40.3 | 30.8  |
> | 0.1      | 37.0       | 37.4 | 25.6  | 27.0       | 65.6 | 7.4   | 29.9           | 41.3 | 28.8  |
> | 1        | 27.4       | 35.7 | 36.9  | 21.6       | 67.2 | 11.2  | 23.7           | 43.0 | 33.3  |
>
> The above illustrates the performance variation of our proposed method as parameter A ranges from $0.0$1 to $1$.  Experimental results are obtained by comparing our method with the PPO with KL penalty in terms of the win/tie/lose ratio.  It can be observed that when the hyperparameter falls within the range of $0.01$ to $0.1$, our proposed method still outperforms the PPO algorithm. However, when the hyperparameter exceeds $0.1$, the model performs poorly.
>
> We **have included** the above analysis and results in **Appendix B.2**.
>
> ---
>
> [1]  Arjovsky, M., Bottou, L., Gulrajani, I., & Lopez-Paz, D. (2019). Invariant risk minimization. arXiv preprint arXiv:1907.02893.
>
> [2] Creager, E., Jacobsen, J. H., & Zemel, R. (2021, July). Environment inference for invariant learning. In International Conference on Machine Learning (pp. 2189-2200). PMLR.
>
> [3] Chen, Y., Xiong, R., Ma, Z. M., & Lan, Y. (2022). When Does Group Invariant Learning Survive Spurious Correlations?. *Advances in Neural Information Processing Systems*, *35*, 7038-7051.
>
> [4] Lin, Y., Zhu, S., Tan, L., & Cui, P. (2022). ZIN: When and How to Learn Invariance Without Environment Partition?. *Advances in Neural Information Processing Systems*, *35*, 24529-24542.

---

> > ### Comment · Reviewer_bjuC · 2023-11-23
> >
> > Thank the author for conducting the experiments to address my concerns. My concerns are mostly addressed. If there is a 7 score I’d be happy to raise it to. But score 8 maybe a big jump. So I’ll keep my score.

---

> > > ### Author Response · Authors · 2023-11-23
> > > **Official Comment by Authors**
> > >
> > > We hope this response addresses your concerns and provides a clearer perspective on the motivations and aspirations of our study. If you find our clarifications satisfactory, we kindly ask for your consideration in adjusting your scores accordingly. Your insights are valuable to us, and we appreciate your time and attention to our work. :-D

---

### Official Review · Reviewer_cgwB · 2023-11-02

**Soundness:** 3 good
**Presentation:** 3 good
**Contribution:** 3 good
**Rating:** 6
**Confidence:** 3

**Summary:**

The paper presents an innovative method addressing the challenge of 'reward hacking' and neglect of complex samples in AI assistants powered by language models, through Reinforcement Learning from Human Feedback (RLHF). The proposed technique advances a consistent policy learning across different data groups, enhancing the AI’s performance evenly across domains. It innovatively classifies data into groups to highlight performance variances, optimizes policy for difficult groups, and adaptively regulates exploration space, thus boosting training stability and generalization.

**Strengths:**

1. **Organization and Accessibility**: The paper is well-structured and the content is presented in a manner that is accessible to readers.

2. **Significance of Addressed Problems**: The authors tackle critical issues within RLHF, such as reward hacking and the overlooking of complex samples, which are pertinent for the advancement of universal AI assistants.

3. **Innovative Concept**: The application of group-invariant learning to the alignment problem is both a novel and a promising idea.

**Weaknesses:**

1. The explanation in Section 4.4 regarding the probability of assigning samples to the highest-performing group lacks clarity and warrants further detail.

2. The term $R_{g}(\theta)$ is not clearly defined within the paper. It’s assumed to represent the expected return of group $g$, yet its relationship to the last term in Equation 6 is ambiguous and needs clarification.

3. The first term of the final learning objective in Equation (8) does not appear to be directly related to group $g$, raising questions about its role in achieving invariant learning.

4. The paper would benefit from the inclusion of code or detailed pseudocode to clearly convey the training process, as the current description does not sufficiently outline the methodology.

5. The connection between group invariant learning and its ability to address issues of shortcut exploitation and neglect of challenging samples is not clearly articulated. More intuitive explanations or empirical evidence would strengthen the understanding of this relationship.

**Questions:**

Please refer to weaknesses

---

> ### Author Response · Authors · 2023-11-21
> **Official Comment by Authors**
>
> Thank you for your valuable comments! Based on your feedback, we have revised the statements in the **method** section to make them more reader-friendly and easier to understand. **The changes are marked in blue.**
>
> **Q1: The explanation in Section 4.4 regarding the probability of assigning samples to the highest-performing group lacks clarity and warrants further detail.**
>
> A1: To enhance the clarity of the explanation in Section 4.4, we **have revised** the relevant context.
>
> 1) In our method, each trajectory $\tau={x, y}$ is assessed by a group inference classifier $\phi$ that is trained to predict its likelihood of belonging to the highest-performing group, denoted as $p_\phi(g_{\mathrm{high}}|\tau)$.
> 2) For data in the **highest-performing group**, we apply a **larger penalty** $\eta \cdot p_\phi(g_{\mathrm{high}}|x,y) $ to prevent reward hacking [1]. This means we avoid excessively favoring data that already shows good performance. On the other hand, for data that are harder to optimize, which have a lower probability of being in the best group $p_\phi(g_{\mathrm{high}}|x,y)$, we relax their constraints. This increases the exploration space for the model. The aim here is to encourage the model to explore and learn from data that are not as easily optimized.Through this method, our approach strikes a balance between exploration and training stability.
>
> **Q2: The term $R_g(\theta)$ is not clearly defined within the paper. It’s assumed to represent the expected return of group $g$, yet its relationship to the last term in Equation 6 is ambiguous and needs clarification.**
>
> A2: We apologize that our definition and notation have mislead you. To clarify, the return $R_g(\theta)$ of a specific group $g$ is a key concept that depends on the group labels within our dataset. To denote whether a particular trajectory $\tau_i$ belongs to group $g$, we use the indicator function $\mathrm{1}\{g_{\tau_i} = g\}$. The return for each group $g$ can then be mathematically expressed as follows:
>
> $R_g(\theta) = \frac{1}{\sum_{i'} \mathrm{1}\{g_{\tau_{i'}} = g\}} \sum_i \mathrm{1}\{g_{\tau_i} = g\}  \left[ \sum_{t=1}^T  \log \pi_\theta(a_t|s_t) R(s_t, a_t) \right]$.
>
> The **last term** within the brackets represents the **expected return along a specific trajectory**. This expected return is then **averaged across all trajectories in group $g$**, providing the **expected return for the group $g$**.
>
> In light of your feedback, we **have revised** the manuscript to include this clarification, ensuring that the definition and the computational method of $\ R_g(\theta)$ are clearly articulated.
>
> **Q3: The first term of the final learning objective in Equation (8) does not appear to be directly related to group $g$, raising questions about its role in achieving invariant learning.**
>
> A3: Thank you for pointing it out.
>
> The **first term** in Equation (8) represents the **standard objective of policy gradient methods**, which is to maximize the expected sum of logged probabilities of actions taken, weighted by their respective returns. This term is fundamental to the reinforcement learning (RL) aspect of the model, aiming to optimize the policy $\theta$ to maximize expected returns.
>
> In the context of invariant learning, the objective is to optimize the policy in a way that is sensitive to and maintains performance across different groups $g$ in the dataset. While the first term of Equation (8) does not directly reference these groups, it serves as a **baseline objective** for the policy optimization. The **group-specific** considerations are introduced through the regularization term $R_{var}(\theta, \phi) $, which is weighted by $ \beta_{\mathrm{policy}}$. This regularization term is designed to ensure that the learned policy performs consistently across different groups, thereby promoting invariant learning.
>
> To make this connection clearer in the manuscript, we **have made revision** to the relevant section.
>
> **Q4: The paper would benefit from the inclusion of code or detailed pseudocode to clearly convey the training process, as the current description does not sufficiently outline the methodology.**
>
> A4: Thank you for your valuable feedback. As your suggestion, we have included detailed pseudocode in **Appendix A** to better convey the training process of our proposed policy invariant learning framework.

---

> ### Author Response · Authors · 2023-11-21
> **Official Comment by Authors**
>
> **Q5: The connection between group invariant learning and its ability to address issues of shortcut exploitation and neglect of challenging samples is not clearly articulated. More intuitive explanations or empirical evidence would strengthen the understanding of this relationship.**
>
> A5: Thank you for highlighting the need for a clearer explanation of how group invariant learning addresses issues like shortcut exploitation and the neglect of challenging samples.
>
> 1) Group invariant learning aims to identify and leverage the underlying **causal relationships** within the data, rather than relying on superficial correlations or easy-to-learn patterns (often referred to as shortcuts) [2, 3]. This approach is particularly effective in **mitigating shortcut exploitation**, where models may overfit to non-generalizable or trivial features present in the training data. By focusing on invariant features across different groups, the model is encouraged to learn more robust and generalizable patterns that are true to the underlying causal structure of the data.
>
> 2) Regarding the **neglect of challenging sample**s, invariant learning inherently promotes a more balanced learning process. It ensures that the model performs **consistently** across various data distributions, including those that contain **harder-to-learn or less frequent patterns**. This approach counters the model's natural tendency to overlook challenging samples in favor of easier ones.
>
> 3) Additionally, the concept of Distributionally Robust Optimization (DRO) [4, 5] aligns closely with group invariant learning. DRO has been empirically shown to enhance the **balance and fairness** of outcomes in machine learning models, further supporting the effectiveness of this approach.
>
> We **have revised** the content in Section 4.2 to provide additional context.
>
> ---
> [1] Laidlaw, C., Singhal, S., & Dragan, A. (2023, July). Preventing reward hacking with occupancy measure regularization. In ICML Workshop on New Frontiers in Learning, Control, and Dynamical Systems.
>
> [2] Arjovsky, M., Bottou, L., Gulrajani, I., & Lopez-Paz, D. (2019). Invariant Risk Minimization. arXiv:1907.02893.
>
> [3] Krueger, D., Caballero, E., Jacobsen, J., Zhang, A., Binas, J., Zhang, D., ... & Courville, A. (2021). Out-of-Distribution Generalization via Risk Extrapolation (REx). Proceedings of the 38th International Conference on Machine Learning, PMLR 139.
>
> [4] Levy, D., Carmon, Y., Duchi, J. C., & Sidford, A. (2020). Large-scale methods for distributionally robust optimization. Advances in Neural Information Processing Systems, 33, 8847-8860.
>
> [5] Sagawa, S., Koh, P. W., Hashimoto, T. B., & Liang, P. (2020). Distributionally Robust Neural Networks for Group Shifts: On the Importance of Regularization for Worst-Case Generalization. arXiv:1911.08731.

---

> > ### Comment · Reviewer_cgwB · 2023-11-22
> >
> > Thanks for the author's feedback, I think my concerns are addressed, so I will raise my score. Additionally, I suggest that you can release the code after acceptance if possible, because an official implementation can help readers understand your method directly.

---

### Meta-Review · Area_Chair_ujJg · 2023-12-03

**Metareview:**

The paper proposes methods to improve reinforcement learning from human feedback. Instead of treating all samples equally, the authors propose to separate data into groups and make sure the learning performance is close across groups, instead of over-optimizing on simple data. The authors demonstrate that this approach substantially the generalizability of learning.

Overall there is a consensus among reviewers that the paper is addressing a well-motivated question and proposing a novel approach, and  the results are overall convincing.  There are various clarification questions and presentation suggestions from reviewers, and the authors have provided mostly reasonable responses. We encourage the authors to incorporate the comments and suggestions from reviewers into the final version of their paper.

**Justification For Why Not Higher Score:**

This is a solid paper, and I wouldn't object to promoting the paper to an oral. Though the current reviews might not be enthusiastic enough.

**Justification For Why Not Lower Score:**

This is a solid paper, with nice and intuitive ideas and good execution. The topic is of interest to the broad community, and the reviews are overall very positive.

---

### Decision · Program_Chairs · 2024-01-16

Accept (spotlight)